# The Role of the Tumor Microenvironment in Triple-Positive Breast Cancer Progression and Therapeutic Resistance

**DOI:** 10.3390/cancers15225493

**Published:** 2023-11-20

**Authors:** Qian Pu, Haidong Gao

**Affiliations:** 1Department of Breast Surgery, Qilu Hospital (Qingdao), Cheeloo College of Medicine, Shandong University, Qingdao 266035, China; angeljogging@163.com; 2Oncology Laboratory, Qilu Hospital (Qingdao), Cheeloo College of Medicine, Shandong University, Qingdao 266035, China

**Keywords:** triple-positive breast cancer, tumor microenvironment, occurrence and development, treatment resistance

## Abstract

**Simple Summary:**

This is an important point of view that the development of breast carcinogenesis is not only related to the intrinsic characteristics of the cancer cells, but that it is also strongly dependent on surrounding microenvironmental factors. It is important to note that triple-positive breast cancer accounts for about 10–15% of all breast cancers, and research on the tumor microenvironment for this subtype is still limited. In this review, authors explored the relationship between critical cellular components and factors in the triple-positive breast cancer microenvironment and the inception, advancement, and therapeutic resistance of breast cancer to provide perspectives on the latest research on triple-positive breast cancer. The authors sufficiently described the state of the art and reported the fundamental concepts, allowing an easy comprehension of the intricate dynamics within tumor microenvironment in triple-positive breast cancer.

**Abstract:**

Breast cancer (BRCA) is a highly heterogeneous systemic disease. It is ranked first globally in the incidence of new cancer cases and has emerged as the primary cause of cancer-related death among females. Among the distinct subtypes of BRCA, triple-positive breast cancer (TPBC) has been associated with increased metastasis and invasiveness, exhibiting greater resistance to endocrine therapy involving trastuzumab. It is now understood that invasion, metastasis, and treatment resistance associated with BRCA progression are not exclusively due to breast tumor cells but are from the intricate interplay between BRCA and its tumor microenvironment (TME). Accordingly, understanding the pathogenesis and evolution of the TPBC microenvironment demands a comprehensive approach. Moreover, addressing BRCA treatment necessitates a holistic consideration of the TME, bearing significant implications for identifying novel targets for anticancer interventions. This review expounds on the relationship between critical cellular components and factors in the TPBC microenvironment and the inception, advancement, and therapeutic resistance of breast cancer to provide perspectives on the latest research on TPBC.

## 1. Introduction

In recent years, breast cancer (BRCA) has become the predominant cause of mortality among women in society, surpassing the global prevalence and death toll of lung cancer [1,2]. It is essential to acknowledge that BRCA is not a single disease but rather a malignancy exhibiting significant heterogeneity. BRCA can be classified into four distinct subtypes based on the immunohistochemical identification of the estrogen receptor (ER), progesterone receptor (PR), human epidermal growth factor receptor 2 (HER2), and Ki-67 index: Luminal A, Luminal B, triple-negative, and HER2 overexpression type [3]. Substantial progress has been made in the clinical diagnosis and treatment of BRCA through molecular categorization. However, due to individual patient heterogeneity and the substantial clinical diversity of BRCA, the prognostic value of BRCA has fallen short of expectations [4,5,6,7]. Research has revealed that triple-positive breast cancer (TPBC), representing a unique hormone receptor-positive subtype within BRCA, exhibits positivity for ER, PR, and HER2 [8], with malignancy levels on par with or even exceeding triple-negative breast cancer (TNBC). There is an increasing consensus suggesting that TPBC is associated with higher rates of recurrence, particularly involving internal organs and bone tissue, thus accounting for the poor prognoses [9,10,11]. While research on BRCA has grown over the years, studies focused on TPBC remain scarce, with even fewer reports from China. Therefore, a deeper exploration and analysis of TPBC’s attributes could offer a scientific foundation for clinical treatments.

TPBC accounts for approximately 10% to 15% of all breast cancers [12,13]. Its clinicopathological characteristics are predominantly characterized by higher tumor grade, larger volume, inferior prognosis, increased invasiveness, significant vasculaturization or nerve invasion, and a higher likelihood of axillary lymph node and distant metastases, with bone metastases being mostly prevalent [14,15]. TPBC is defined by the positive expression of ER, PR, and HER2. ER and PR, intracellular receptors binding to estrogen and progesterone, respectively, regulate factor expression and cell proliferation. HER2, a cell surface receptor, stimulates cell proliferation via downstream signaling pathways. Elevated expression of these molecules suggests TPBC’s increased dependence on these receptors and hormones, often prompting anti-HER2-targeted molecular therapies. Clinical treatment typically encompasses comprehensive approaches such as surgical resection, targeted therapy combined with chemotherapy, endocrine therapy, and radiotherapy. Over the years, studies have uncovered that TPBC patients exhibit limited responses to chemotherapy and hormone therapy, possibly due to the cross-interaction between the HR and HER2 signaling pathways. When one pathway is blocked by chemotherapy or hormone therapy, the other is subsequently upregulated [16]. Mark Pegram et al. pointed out that blocking multiple HER family receptors may be more effective than blocking the HER 2 kinase alone, and using pan-HER TKI neritinib as part of the combination therapy may re-sensitize the ER pathway to endocrine therapy [17]. Consequently, treating TPBC warrants a comprehensive consideration of these dual pathways. Precisely evaluating TPBC prognosis and effectively monitoring and predicting tumor recurrence stand as pivotal topics in the era of precision medicine. Assessing patient prognosis based on clinical tumor characteristics offers valuable guidance for clinicians overseeing recurrence monitoring.

In the realm of malignant tumors, invasion and metastasis represent two crucial facets, with metastasis holding greater influence over clinical decision-making, embodying the predominant cause of death in cancer patients [18]. The metastatic course of cancer cells encompasses dissemination from the primary tumor, traversing epithelial-to-mesenchymal transition (EMT), followed by colonization at secondary sites through mesenchymal-to-epithelial transition (MET).

The tumor microenvironment (TME) refers to the evolving microecosystem accompanying normal cells and supporting them during their transformation from normal to malignant states. This localized microenvironment evolves during the tumor’s developmental course [19,20]. The TME encapsulates cellular and non-cellular constituents alongside latent malignant tumor cells [21]. It is widely thought that a bidirectional signal exchange and mutual influence is present, intricately linked to tumorigenesis, metastasis, and treatment resistance [22,23]. Cell components span mesenchymal and hematopoietic cells, the former encompassing fibroblasts, myofibroblasts, adipocytes, endothelial cells, and mesenchymal stem cells (MSCs). Hematopoietic cells involve lymphoid cells (T cells, B cells, dendritic cells, mast cells, NK cells) and bone marrow cells (tumor-associated macrophages, neutrophils, myeloid-derived suppressor cells (MDSCs)). Non-cellular components encompass tangible and intangible elements. Tangible components largely constitute critical extracellular matrix (ECM) constituents such as collagen, fibronectin, laminin, aminoglycans, and proteoglycans. Intangible components predominantly encompass soluble agents like growth factors, cytokines, and chemokines. Additionally, hormones, proteases, and exosomes, as well as features like hypoxia and metabolic imbalances, are present. The tumor microenvironment additionally encompasses pivotal metabolic conditions (pH, PO_2_, glucose, glutamine, lactic acid) and chemical factors (e.g., NO) [7]. Interactions among these TME components collectively promote tumor initiation, development, invasion, and metastasis, and influence responses to treatment. Following the malignant shift of mammary epithelial cells, the microenvironment composition undergoes significant change, restructuring the ECM and releasing constraints on mammary epithelial cell proliferation, consequently leading to uncontrolled tumor cell proliferation and invasion [24]. Drug resistance exhibited by TPBC toward endocrine therapy results from multifaceted mechanisms and signaling pathway coordination, which are not exclusively related to tumor cell transformations, with many TME components playing pivotal roles [25]. Although the past few years have witnessed extensive research on the TME of TNBC, research on TPBC’s TME remains limited. This paper thus reviews the roles of several crucial cellular and non-cellular constituents in TPBC’s onset, progression, and drug resistance pathways via an exploration of related cell line TMEs.

## 2. Tumor Mesenchymal Cell Components in TPBC

### 2.1. Cancer-Associated Fibroblasts (CAFs)

Within the breast cancer TME, CAFs dominate as the most abundant cellular constituents of the tumor stroma. They can originate from normal stromal fibroblasts (referred to as normal fibroblasts, NFs) [26]. Additionally, CAFs can emerge through epithelial-mesenchymal transition from bone marrow-derived mesenchymal stem cells, endothelial cells, epithelial cells, or transdifferentiated cells [27,28]. The widely utilized marker for identifying CAFs is the α-smooth muscle actin (α-SMA) [29]. CAFs secrete growth factors, cytokines, proteases, and more, which not only foster the initiation, growth, angiogenesis, invasion, and metastasis of breast cancer but also serve as biomarkers in clinical practice, guiding breast cancer diagnosis, therapy, and prognosis assessments [19,30,31,32,33].

In triple-positive breast cancer, Shekhar et al. revealed through a three-dimensional cell–cell interaction model that normal mesenchymal fibroblasts (NFs) inhibited estrogen-induced tumor cell growth, whereas CAFs produced abundant estrogens, inducing the malignant transformation of the normal breast epithelial cell line MCF10A and the precancerous breast epithelial cell line EIII8 [34]. Notably, cell surface antigens like CD44, CD24, and ESA have been effectively harnessed to isolate cancer stem cell (CSC)-like populations from breast cancer cell lines and primary tissues. CD44^hi^ and CD44^lo^ populations have been identified across various basic breast cancer systems, while luminal breast cancer systems exclusively consisted of CD44^lo^ populations. ZEB1 played a crucial role in steering the transition from CD44^lo^ to CD44^hi^. Importantly, CAFs’ secretion of transforming growth factor-beta (TGFβ) enhanced the conversion rate of CD44^lo^ to CD44^hi^ in the basal breast cancer CD44^lo^ cell subset. This effect was contingent on the induction of ZEB1 expression, indicating the considerable role of CAFs in triple-negative breast cancer initiation, although the involvement of CAFs in TPBC initiation has received limited attention [35].

It is well-established that CAFs can release various factors that stimulate neighboring malignant cells, thereby fostering tumor progression [36]. Growth factors secreted by CAFs in the breast cancer microenvironment encompass basic fibroblast growth factor 2 (FGF2) [37], platelet-derived growth factor (PDGF) [38], insulin growth factor 1 (IGF1), and large amounts of C-X-C motif chemokine ligand 12 (CXCL12), also known as stromal cell-derived factor (SDF-1). Interestingly, heightened CXCL12 expression levels could predict diminished overall survival (OS) and relapse-free survival (RFS) in patients across all breast cancer subtypes. CXCL12 is pivotal in the initial stages of tumorigenesis, and studies have corroborated that fibroblast-derived CXCL12 consumption mitigates tumor cell proliferation while recruiting endothelial progenitor cells to the TME, thus expanding tumor vessels [39]. Furthermore, CAFs can partake in mediating inflammation-driven tumor-promoting factors by producing pro-inflammatory cytokines [40,41]. Recent findings have indicated the expression of the third estrogen receptor, the G protein-coupled estrogen receptor (GPER), in breast CAFs. In estrogen-sensitive breast cancer, GPER was found to mediate a distinct gene signature associated with cell growth, migration, and angiogenesis [42,43,44]. E2 and G-1 induced IL1β expression in CAFs through GPER activation. This process also triggered the EGFR/ERK/PKC signaling cascade, culminating in the establishment of a feedforward loop. This loop further interconnected IL1β with the expression of IL1R1 in luminal cancer cells MCF7 and SKBR3, fostering functional crosstalk between the tumor microenvironment and intracavitary breast cancer cells, consequently promoting tumor metastasis [45]. Notably, it has been shown that CAFs secrete exosome miR-500a-5p, which could bind to ubiquitin-specific peptidase 28 (USP28), thereby facilitating progression and metastasis in both the basal breast cancer cell line MDA-MB-231 and the luminal cell line MCF7 [46].

Both endocrine therapy and targeted therapy are indicated for TPBC. However, the involvement of CAFs in the emergence of resistance to anticancer drugs remains a complex area, particularly regarding the specifics of CAF-mediated tamoxifen resistance in TPBC. Investigations have unveiled that interleukin-6 (IL-6) secreted by CAFs contributes to the downregulation of ERα expression and activates downstream signaling pathways, notably JAK/STAT3 and PI3K/AKT. This orchestration results in resistance to endocrine therapy, especially in luminal cell carcinoma cells [47]. The emergence of tamoxifen (TAM) resistance stands as a significant hurdle in endocrine therapy for BRCA. The G-protein-coupled estrogen receptor (GPER) has been postulated to be a catalyst for tamoxifen resistance. Current evidence suggests that TAM, alongside E2 and GPER agonist G1, stimulates proliferation, cell-cycle progression, and E2 production through the GPER/EGFR/ERK axis within breast CAFs. These findings introduce novel perspectives into the GPER-mediated, CAF-dependent mechanism underlying TAM resistance in breast cancer [48]. Furthermore, GPER was identified as influencing β1-integrin, a target gene, wherein CAF’s release fibronectin (FN), which engages with β1 integrin, thereby activating MAPK/ERK1/2 and PI3K/AKT pathways. The resultant cascade contributes to endocrine therapy drug resistance in breast cancer [49]. Yuan et al. further found that GPER/EGFR/ERK signaling boosts β1-integrin expression and triggers downstream kinases in MCF-7R cells, fostering CAF-induced cell migration and epithelial–mesenchymal transition (EMT). GPER’s potential to promote tamoxifen resistance is linked with its interaction within the tumor microenvironment in a β1-integrin-dependent manner [50].

It is widely thought that anti-HER2-targeted therapy is pivotal in the context of triple-positive breast cancer treatment. However, the mechanism behind how cancer-associated fibroblasts induce trastuzumab resistance in HER2^+^ breast cancer remains elusive. Current understanding proposes that CAFs in HER2^+^ breast cancer confer resistance to Herceptin by orchestrating the loss of PTEN or activating the IL-6/STAT-3/NF-κB signaling pathway [51]. Furthermore, CAFs can trigger trastuzumab resistance by expanding tumor stem cells and activating multiple pathways, such as NF-κB, JAK/STAT3, and PI3K/AKT. In this light, a combined approach featuring anti-IL-6 antibodies or multi-pathway inhibitors in conjunction with trastuzumab could emerge as a novel strategy to counteract trastuzumab resistance [52].

### 2.2. Tumor-Associated Macrophages (TAMs)

Immune cells within the TME wield a dual influence, inhibiting tumors while also participating in processes like tumor angiogenesis, immunosuppression, and cell growth through cytokine secretion. This multifaceted interplay often fosters tumor metastasis and bolsters tolerance to drug treatments [53]. Among the immune cell populations, macrophages, mast cells, and lymphocytes are the primary inflammatory cells involved in the TME.

TAMs constitute one of the most prevalent immune cell subsets within the TME and significantly shape various aspects of solid malignancies. They catalyze tumorigenesis, promote neovascularization, remodel the TME into an immunosuppressive state, confer resistance to cancer chemotherapy, and increase the risk of recurrence and metastasis [54]. TAMs can be categorized into two functionally disparate subtypes: classically activated M1 TAMs, which possess anti-tumor traits, and activated M2 TAMs, which potentiate tumor cell formation and metastasis [55]. An increased density of M2 phenotype TAMs is closely associated with unfavorable clinical outcomes [56]. Typical markers of M1 macrophages encompass CD64, CD68, CD80, CD86, IDO, SOCS1, and CXCL10 [57], while M2 macrophages are typified by markers like MRC1, TGM2, CD23, CD163, CD204, CD206, and CCL22 [58,59,60]. In a meta-analysis involving 4541 breast cancer patients, Matikas et al. uncovered that elevated TAM density within tumor tissue correlated with poorer prognosis and hormone-receptor-negative (HR-) status [61]. In a study by Pelekanou et al. involving 113 HR+ and 37 ER- patients who underwent tumor biopsies at baseline and two cycles of neoadjuvant chemotherapy (NAC), gene mapping analysis and quantification of tumor-infiltrating lymphocytes (TILs), FOXP3+ T cells, and CD163+ TAMs, were conducted. The study unveiled that heightened CD163 expression correlates with enhanced OS in ER+ patients [62]. Additionally, Mansfield et al. were among the first to propose that reduced CD163+ TAMs may play a role in lymph node metastasis in breast cancer, with CD163 expression serving as a predictor of lymph node status [63]. Additionally, studies by FAN and He indicated that in the luminal breast cancer TME, TAM characteristics include a lower quantity, suppression, rounded morphology, and slight elevation in M2 markers [64]. Discrepancies in the relationship between TAM markers and pathologic complete response (pCR) expression and prognostic indicators could arise due to inconsistent immunohistochemical interpretation of TAMs or be associated with the molecular subtype of breast cancer. Accordingly, the predictive value of TAMs for treatment response across different BRCA subtypes, especially in TPBC, warrants further investigation [65].

While the relationship between TAMs and TNBC has become a research hotspot, studies examining TAMs and TPBC remain scarce. Interestingly, it has been reported macrophages can drive the development of triple-negative tumors by regulating succinate dehydrogenase (SDH) expression in tumor cells through mechanisms linked to the TGF-β signal transduction [66]. The overexpression of MCT-1 in triple-negative breast cancer activates MCT-1/miR-34a/IL-6/IL-6R signaling, disrupting the normal structure of breast cancer epithelial glands and inducing EMT, thus promoting the polarization of TAMs toward the M2 subtype [67]. Additionally, tumor heterogeneity significantly influences the activation of tumor-associated macrophages. Maija et al. employed full transcriptomic sequencing to study human monocytes co-cultured with ER+ or TNBC cell lines. They investigated the biological responses associated with differential gene activation in monocytes and cancer cells, revealing divergent macrophage phenotypes induced by ER^+^ (T47d) and TNBC (MDA-MB-231) cancer cells, exhibiting distinct functions, cytokine and chemokine secretion patterns, and morphology. Simultaneously, the presence of macrophages exerted considerable effects on ER+ and TNBC cell lines. ER^+^ cells demonstrated enhanced acute inflammatory responses, IL-17 signaling, and antigen-presenting pathways, while thioredoxin and vitamin D3 receptor pathways were downregulated in macrophages [68]. Derived from TAMs, the CXC motif chemokine ligand 1 (CXCL1) has been demonstrated to drive the differentiation of naive CD4+ T cells into regulatory T cells through transcriptional activation of the NF-κB/FOXP3 signaling pathway, contributing to the development of regulatory T cells (Treg) [69]. Similarly, mediating CXC motif chemokine ligand 8 (CXCL8) infiltration fosters immune escape and lung metastasis in BRCA [70].

In recent years, TAMs have been linked to drug resistance in specific solid tumor types, including BRCA. Nonetheless, investigations into the interplay between TAMs and therapeutic resistance in TPBC are in their preliminary stages. TAMs are widely believed to induce drug resistance via the IL-10/STAT3/bcl-2 signaling pathway [71]. Additionally, the secretion of chemokine ligand 2 (CCL2) by TAMs triggers the activation of the PI3K/AKT/mTOR signaling cascade in tumor cells, culminating in tamoxifen resistance within breast cancer [72]. TAMs can also activate the NF-κB/STAT3/ERK pathway within tumor cells through the release of pro-inflammatory cytokines, such as TNF-α and IL-6. This signaling cascade phosphorylates and activates ERα, leading to endocrine therapy resistance in BRCA [73]. M2-type TAMs have demonstrated the ability to activate breast cancer cells via the EGFR/PI3K/Akt pathway, resulting in the upregulation of SGLT1, a phenomenon that fosters tamoxifen resistance in cancer cells [74]. Notably, M2 macrophages play a pivotal role in instigating drug resistance in BRCA, with exosomes also assuming a crucial role in the induction process of M2 macrophages. An exosome-mediated release of miR-222 by adriamycin-resistant MCF-7 cells facilitates M2 polarization through the PTEN/Akt pathway, thereby accelerating tumor progression [75].

### 2.3. Cancer-Associated Adipocytes (CAAs)

Adipose tissue is an important component of the mammary gland, accounting for 56% [76] and 35% [77] of breast volume during non-lactating and lactating periods, respectively. Bioactive adipokines emanating from adipocytes have been identified as major contributors to tumorigenesis and cancer progression [78]. CAAs may promote proliferation, migration, and invasion of BRCA cells [79], drive metabolic alterations within breast cancer cells, induce heightened fatty acid storage and oxidation [80], and mediate tumor resistance to chemotherapy and radiation therapy [81]. CAAs may even be associated with an increased incidence of local events after autologous adipose tissue filling for breast cancer [82]. Additionally, CAAs are widely thought to play an important role in the development of TPBC.

Pathologically, CAAs are associated with the formation of crown-like structures (CLS) and enhanced lipolysis. These CLS—microscopic conglomerates of expiring adipocytes encircled by macrophages—are believed to reflect sites with heightened conversion of androgens to estrogens [83]. CLS constitute a histological hallmark of breast adipose tissue, marked by organized assemblies of CD68-positive macrophages enveloping deceased or perishing adipocytes. It is highly conceivable that these structures indicate chronic inflammation. Studies involving animal models and female breast tissue have suggested that CLS’ presence is correlated with obesity and pro-inflammatory and pro-carcinogenic processes, marked by augmented aromatase activity and signaling via an ER-mediated pathway [84,85,86]. It has also been reported that CLS adversely affect ER-positive breast cancer prognosis. In mice, the CLS-associated pro-inflammatory and carcinogenic processes can be ameliorated through weight loss or chemopreventive drugs [87,88]. Mullooly et al. [89] conducted a study that explored the relationship between CLS and sex steroid hormones in breast adipose tissue and serum within 83 postmenopausal breast cancer patients. They discovered that the presence of CLS in the breast adipose tissue of these patients was linked to an increased estrogen-to-androgen ratio in both breast fat and serum. Cha YJ et al. [90] compared breast adipose tissue among normal individuals, breast cancer patients, and adipose tissue adjacent to breast cancer. They observed the densest distribution of CLS in adipose tissue adjacent to cancer (*p* < 0.001) was correlated with age, ER expression, and HER2 expression.

In contrast to normal adipose tissue, CAAs exhibit diminished differentiation and lose their typical functions, including energy storage [91]. They also secrete a plethora of soluble cytokines, referred to as adipokines/adipocytokines [92], with ongoing research focusing on factors like leptin, adiponectin, visfatin, autotaxin, hepatocyte growth factor (HGF), tumor necrosis factor-alpha (TNF-α), and heparin-binding epidermal growth factor (HB-EGF) [93].

Regarding TPBC, CAAs could potentially activate several signaling pathways, including ER signaling. Leptin, when lacking a natural ligand, stimulates ER signaling, ultimately propelling BRCA cell proliferation [94,95]. Furthermore, leptin enhances the cellular proliferation of BRCA cells through HER2 trans-activation via EGFR and JAK2 activation [96,97]. Intriguingly, leptin’s influence even extends to increased cellular proliferation facilitated by the JAK/STAT3 [98] and PI3K/AKT pathways [99]. Notably, within luminal cell lines MCF-7 and SK-BR-3, leptin can induce EMT in BRCA cells by upregulating PKM2 expression and activating the PI3K/AKT signaling pathway. Counteracting the effects of leptin through an anti-leptin receptor antibody (anti-OBR) and PI3K/AKT signaling inhibitor LY294002 significantly suppresses leptin-induced PKM2 expression, as well as the expression of EMT-related markers [100]. Partial inhibition of leptin-induced migration, invasion, and EMT-related marker expression can be achieved through siRNA targeting PKM2 [100].

Adiponectin, a hormone predominantly synthesized in adipose tissue, is abundantly present in human plasma and has been recognized as an insulin-sensitizing adipocytokine [101,102]. Gunter et al. [103] conducted a retrospective analysis involving 875 postmenopausal breast cancer patients and 839 volunteers and found significantly reduced levels of adiponectin in the blood of breast cancer patients, concurrent with a decrease in adiponectin levels in CAAs [104,105]. AdipoR1 and AdipoR2, two transmembrane adiponectin receptor subtypes, initiate adiponectin’s biological effects by inducing the activation of protein kinases, primarily AMPK and MAPK [106]. In luminal cells, estrogen—the primary mitogen of breast cancer cells—has been reported to induce G1-S cell cycle progression by upregulating the expression of c-myc and cyclin D1 genes [107]. Dieudonne MN et al. identified functional AdipoR1 and AdipoR2 receptors in MCF-7 cells and demonstrated that adiponectin suppressed the expression of c-myc and cyclin D1 mRNA, directly impeding the cell cycle progression in the G1-S phase. Adiponectin exerted anti-proliferation and pro-apoptotic effects through the activation of AMP kinase and MAP kinase pathways, including p38 MAPK, ERK, and JNK [108]. Concurrently, adiponectin upregulated the expression of the tumor suppressor gene LKB1 in MCF7 cells. LKB1 was found to be crucial for adiponectin-mediated regulation of the AMPK-S6K axis, which mitigated adhesion, migration, and invasion of breast cancer cells [109].

Current evidence suggests that within crown-like structures, dying CAAs liberate free fatty acids, binding to toll-like receptor 4 on macrophages and adjacent adipocytes. This binding triggers the activation of nuclear transcription factor κB, resulting in the upregulation of inflammatory factors, including IL-8. Extracted from invasive breast cancer specimens, CAAs generated significantly more IL-8 compared to neighboring normal fat cells [110]. Elevated IL-8 levels correlated with earlier distant metastasis and poorer breast cancer prognosis [111]. While some scholars advocate that elevated IL-8 is mainly detected in hormone receptor-negative and HER2-positive breast cancer cases [112,113], a complex interplay exists between ER expression and IL8 in BRCA cells [114,115,116]. The underlying mechanism suggests IL-8’s involvement in the activation of intracellular BRCA pathways, such as the phosphatidylinositol 3 kinase/mitogen-activated protein kinase pathway, phospholipase C/protein kinase C pathway, and non-receptor protein tyrosine kinases, including Src family members and adhesion spot kinases [117]. Additionally, cancer cell-secreted IL-8 can convert normal fat cells into CAAs, heightening CAAs’ capacity to stimulate angiogenesis and promote tumor growth [110].

The TME assumes a pivotal role in tumor drug resistance, with CAAs playing a significant role in modulating tumor response to therapy. In ER^+^ breast cancer cells, CAAs secrete fibroblast growth factor 1 (FGF1), which phosphorylates the fibroblast growth factor receptor (FGFR). This process leads to resistance to endocrine therapy in BRCA. However, inhibiting the FGFR could potentially reverse this resistant phenotype [118]. In addition, leptin holds the potential to influence anti-cancer therapy by promoting the proliferation of cancer stem cells [119].

### 2.4. Mesenchymal Stem Cells (MSCs)

MSCs are pluripotent spindle-shaped cells obtained and cultured from diverse tissues like bone marrow, fat, umbilical cord, and placenta [120]. They exhibit surface markers CD73, CD90, and CD105 and lack the expression of CD34, CD45, CD14 or CD11b, CD19 or CD79α, and human leukocyte antigen-DR (HLA-DR). MSCs harbor trilineage differentiation potential (osteogenesis, lipogenesis, and chondrogenesis), enabling them to differentiate into bone, cartilage, muscle, fat, and nerve cells in vitro [121]. The comprehensive mechanism and impact of MSCs from varied sources (such as bone marrow, umbilical cord, fat, and placenta) on triple-positive breast cancer have yet to be systematically investigated. Ongoing research is focused on the role of MSCs from different origins in luminal cell lines and on preliminarily assessing their pro-tumor or anti-tumor effects in TPBC, along with plausible mechanisms underlying treatment resistance.

MSCs can secrete chemokine ligand 5 (CCL5) and transforming growth factor-beta (TGF-β), fostering EMT and preserving stem cell properties within tumor cells, thereby driving metastasis [122]. Breast cancer stem cells (BCSCs), with attributes resembling stem cells, including self-renewal and oncogenic maintenance, drive the proliferation, metastasis, and chemotherapy resistance of breast cancer cells. MSCs participate in the regulation of BCSCs by secreting interleukin-6 (IL-6) and cytokine CXCL7, thereby promoting BCSC self-renewal and cancer cell proliferation [123]. MSC-derived exosomes, rich in mRNA, miRNA, and soluble factors (including cytokines and chemokines), can infiltrate tumor cells, exerting biological effects [124]. These exosomes have been found to inhibit the mTOR/HIF-1α/VEGF signaling axis, thereby suppressing the expression of relevant factors, inhibiting endothelial cell migration and proliferation, and even inducing dormancy in BRCA cells, restraining the migration and proliferation of breast cancer cells [125,126]. In vitro studies focusing on adipose-derived stem cells (ASCs) have demonstrated that ASCs’ extracellular signaling exerts an inhibitory influence on BRCA progression. Under direct co-culture conditions, ASCs notably mitigated the proliferation of SKBR3 cells, an effect dependent on the SDF1α/CXCR4 signaling axis. Concurrently, ASCs induced alterations in tumor cell morphology, prompting epithelial-to-mesenchymal transition, increased the formation of breast spheres, facilitated cell fusion, and augmented the migratory capacity of SKBR3 cells. These outcomes could potentially be associated with the stimulation of cytokines and chemokines secreted by MSCs in BRCA cells [127]. In a recent study, human breast cancer luminal cell line MCF-7 and basal cell line MDA-MB-231 were exposed to a medium containing extracellular vesicles derived from human umbilical cord mesenchymal stem cells (Huc-MSC-EVs), leading to enhanced cell proliferation, migration, and invasion through activation of the ERK pathway. A simultaneous decrease in E-cadherin expression and an increase in N-cadherin expression were observed, promoting the process of epithelial–mesenchymal transition in breast cancer cells [128].

MSCs exert their influence on the breast cancer stem cell pathway by secreting IL-6 and generating factors originating from mesenchymal cells, which may underlie the mechanism of MSC-BCSC mediation in breast cancer drug resistance [129]. The secretion of IL-6 by MSCs contributes to the development of chemotherapy resistance and encourages the proliferation of ERα^+^ breast cancer cells [130]. Notably, IL-6 can induce resistance development in breast cancer cells against cytotoxic drugs intended for ERα^+^ breast cancer (like paclitaxel and doxorubicin) and anti-HER2+ breast cancer treatment, like Herceptin [51].

## 3. Non-Cellular Components in the TME of TPBC

### 3.1. Tumor-Derived Exosomes (TDEs)

Exosomes are nanoscale membrane vesicles released by a variety of cell types and significantly contribute to breast cancer pathogenesis, encompassing tumor initiation, invasion, metastasis, angiogenesis, immune modulation, and the tumor microenvironment. Additionally, they are instrumental in the emergence of drug resistance [131]. Research highlights that exosomes can transport cellular components, including proteins (like cytokines, membrane receptors, and ligands), nucleic acids (such as DNA, mRNA, long and short non-coding RNAs), and lipids, which are capable of influencing other cells or being selectively transferred to recipient cells. This, in turn, affects the behavior and function of these recipient cells [132]. Consequently, they induce alterations in the tumor microenvironment via cytokine, growth factor, chemokine, and MMP secretion, subsequently fostering tumor progression by inducing processes like extracellular matrix remodeling, tumor angiogenesis, immune evasion, and escape [133]. Notably, exosomes can activate the EMT and WNT signaling pathways, which are prevalent regulatory pathways for modifying breast cancer metastatic potential. These exosomes can facilitate breast cancer cell metastasis by inducing EMT through proteins, like MAP17, present in their cargo [134]. Similarly, exosomal miR-146a can reportedly trigger breast cancer cell metastasis by activating the WNT signaling pathway [135]. Furthermore, exosomes released by highly metastatic breast cancer cells can disrupt the blood–brain barrier, thereby facilitating brain metastasis [136]. Recent studies have underscored the potential of exosomal miRNAs as a novel avenue for breast cancer treatment [137], especially against triple-negative breast cancer [138].

In their capacity as adept carriers, exosomes transport molecules, such as miRNAs and proteins, to recipient cells and the tumor microenvironment. Consequently, they directly or indirectly influence cancer cell drug resistance via interactions with the tumor microenvironment. Exosomes enhance breast cancer cell resistance to drugs by potentiating anti-apoptotic capacities, inducing cancer cell autophagy, expediting drug efflux, inducing immunosuppression, or heightening cell stemness. A significant correlation exists between breast cancer metastasis and drug resistance. Cellular exosomes originating from chemotherapy-resistant breast cancer may foster metastasis in cancer cells through the EphA2-Ephrin A1 reverse signaling [139]. Moreover, exosomes from radiation-resistant breast cancer cells, once absorbed by radiation-sensitive cells, can augment the metastatic potential of the latter [140]. This underscores the role of exosomes in both promoting breast cancer metastasis and inducing drug resistance through pathways linked to metastasis.

Importantly, TDEs could potentially contribute to the drug resistance encountered in TPBC. In terms of endocrine therapy, cellular exosomal miRNA-205 could hinder apoptosis in breast cancer cells by blocking the Caspase signaling pathway, thereby heightening the cells’ resistance to tamoxifen [141]. In tamoxifen-resistant breast cancer cells, exosomes released by these cells might negatively regulate ADIPOQ expression in recipient cells, rendering more cancer cells tolerant to chemotherapy drugs upon interaction with surrounding drug-sensitive tumor cells [142]. Regarding targeted therapy, breast cancer serum exosome AGAP2-AS1 has been associated with altered drug resistance patterns in the context of the anti-tumor drug trastuzumab [143,144]. Autophagy, a protective cellular mechanism, is triggered by anticancer drugs in cancer cells [145]. The exosomes released by trastuzumab-resistant breast cancer cells are rich in lncRNA AGAP2-AS1, which activates the autophagy reaction of cancer cells after targeting ATG10 [143]. Importantly, exosomal transport of miR-567 into breast cancer cells can diminish their tolerance to trastuzumab by inhibiting cancer cell autophagy [146].

### 3.2. Extracellular Matrix (ECM)

The extracellular matrix primarily comprises proteins generated by stromal cells like epithelial cells, endothelial cells, and fibrocytes. These proteins modify tissue structure and integrity by arranging components spatially and altering their physical attributes. The ECM exerts its influence not only by virtue of its constituent proteins but also by aiding the action of soluble factors within the tumor microenvironment [49]. ECM stiffening, a form of ECM remodeling, is attributed to collagen accumulation and the cross-linking and aggregation of collagen fibers. This phenomenon has a strong correlation with the progression and clinical prognosis of breast cancer [147,148]. It is widely acknowledged that tumor cells are capable of modulating ECM composition, thereby hindering immune cells and therapeutic agents from accessing tumor cells. This orchestrated effect serves to facilitate immune evasion and confer tolerance to drug treatments. Importantly, the component proteins of the ECM are intricately linked to tamoxifen resistance in breast cancer [149].

### 3.3. Cytokines and Growth Factors

Cytokines and growth factors represent soluble factors within the tumor microenvironment produced by various cell types, including epithelial cells, immune cells, and stromal cells. These molecules engage in signaling interactions between tumor cells and other components within the tumor microenvironment. Consequently, they influence the growth, differentiation, and movement of tumor cells. In the context of ER-positive breast cancer cells, paracrine signaling involving the platelet-derived growth factor (PDGF-CC) stimulates CAFs through binding to the platelet-derived growth factor receptor (PDGFR). This stimulation prompts CAFs to express a hepatocyte growth factor (HGF), an insulin-like growth factor binding protein 3 (IGFBP3), and stanniocalcin1 (STC1), ultimately leading to endocrine therapy resistance. Notably, targeted inhibition of PDGF-CC has the potential to restore tumor cell sensitivity to endocrine therapy, highlighting a potential avenue for intervention [150].

## 4. Discussion

The TME plays a pivotal role throughout all stages of BRCA development [151]. It primarily comprises various cell types, including fibroblasts, macrophages, and endothelial cells. Among these, CAFs stand out as influential entities in the TME, actively releasing cytokines and chemokines that substantially contribute to the growth and invasiveness of cancer cells [152,153]. The administration of chemotherapy drugs triggers tumor cells to secrete diverse cytokines, a process that recruits CAFs [154,155]. In turn, CAFs can foster drug resistance in tumor cells by releasing cellular exosomes [156]. Although CAFs have been implicated in driving drug resistance in breast cancer cells, the precise mechanisms underlying their interactions with tumor cells warrant further scientific investigation. Notably, tumors rich in CAFs have been linked to inferior prognoses compared to those with fewer CAFs [30]. This correlation is particularly pronounced in TNBC, where patients with stroma-rich tumors tend to experience worse overall survival outcomes than those with stroma-poor tumors [157]. Thus, molecules secreted by CAFs, including CXCL14 and CCL5, have emerged as potential therapeutic targets for TNBC treatment [158]. Nevertheless, the role of CAFs within the tumor microenvironment of TPBC remains understudied. The molecular mechanisms underlying how CAFs influence TPBC proliferation, progression, and invasiveness remain unclear, and further exploration is needed to pinpoint potential therapeutic targets for TPBC involving CAFs.

Peritumoral adipocytes undergo regulatory changes instigated by tumor cells, transforming into cancer-associated adipocytes (CAAs). These CAAs subsequently secrete pro-inflammatory cytokines, growth factors, extracellular matrix proteins, and metabolites. These secretions collectively modulate energy metabolism, extracellular matrix remodeling, and immunomodulatory functions within tumor cells, thereby promoting the initiation and progression of breast cancer [159]. Furthermore, exosomes released by adipocytes could expedite the malignant evolution of breast cancer by triggering the Hippo signaling pathway and prompting metabolic reprogramming [160]. In light of these findings, CAAs represent a promising target for breast cancer treatment; however, their practical application remains a distant objective. Future scientific research must address several issues. Firstly, it is crucial to decipher the intricate interactions and specific mechanisms that occur between CAAs and other constituents of the tumor microenvironment, including triple-positive breast cancer cells, tumor-associated fibroblasts, and immune cells. Additionally, it is essential to ascertain whether CAAs predominantly mediate the association between obesity and triple-positive breast cancer.

Non-cellular exosomes within the tumor microenvironment exert a distinct influence on breast cancer cell drug resistance. Cellular exosomes are widely dispersed across various body fluids [161]. Possessing bilayer lipid membranes and nanoscale dimensions, exosomes are adept at evading clearance or damage by cells, such as complements or macrophages. This unique property significantly prolongs the circulating half-life of exosomal contents, enhancing their biological activity [162]. Notably, nanoparticles enveloped in cellular exosome membranes can protect internal siRNA and drugs from degradation, offering the potential for targeted delivery to tumor cells [163,164]. This underscores the potential utility of exosomes and their contents as markers for monitoring drug resistance in breast cancer. Relevant investigations have detected alterations in plasma exosome-associated components among breast tumor patients, offering predictive value for changes in breast tumor cell sensitivity to chemoradiotherapy drugs. Notably, an elevated expression of TK1, CDK9, and CD44 within plasma exosomes from drug-resistant breast cancer patients correlates with therapy resistance [165,166]. Chemoradiotherapy drugs not only prompt the release of exosomes from drug-resistant tumor cells but also elicit exosome release from other cells within the tumor microenvironment. This indirect influence subsequently impacts chemotherapy drug resistance in breast cancer cells. For instance, doxorubicin therapy triggers the upregulated expression of miR-21-5p within exosomes secreted by mesenchymal stem cells, subsequently inducing doxorubicin resistance in breast cancer cells via the microenvironment [165]. Moreover, stimulation of breast cancer cells with paclitaxel enhances their exosome-releasing capacity [167], an effect observed even when breast cancer cells are insensitive to aromatase inhibitor therapy [168]. Furthermore, tumor-released exosomes can modulate the sensitivity of breast cancer cells to drugs by dampening immune cell-mediated inflammatory responses and inducing immunosuppressive cells. Indeed, breast cancer-derived exosomes absorbed by bone marrow cells activate STAT3, leading to CXCR4 reduction. Consequently, bone marrow cells differentiate into bone marrow-derived suppressor cells (MDSCs), inhibiting T lymphocyte proliferation and considerably reducing cancer cell sensitivity to chemotherapy drugs [169]. Moreover, tumor-induced mesenchymal stem cell exosomes can drive MDSC differentiation and foster their transformation into M2-type macrophages endowed with potent immunosuppressive traits, thereby attenuating the anti-tumor effect [170]. Last but not least, radiotherapy further triggers the release of exosomes from tumor cells. As breast cancer cell damage increases, exosome secretion activity and pathways are heightened proportionally to the radiotherapy dose. Importantly, exosomes from drug-treated breast cancer cells may exert an influence on drug resistance in untreated cells within the same group [171]. In essence, exosomes emerge as a critical player in the development of breast cancer drug resistance, offering a valuable avenue for tackling clinical breast cancer treatment resistance. Although research on exosomes in TPBC is limited, further exploration of their mechanisms within the tumor microenvironment is crucial for understanding the progression of drug resistance in TPBC.

## 5. Conclusions and Perspectives

As of 2021, breast cancer has surpassed lung cancer to become the most prevalent cancer globally, with a significant increase in incidence both domestically and internationally [172,173]. Roughly 25% of all breast cancers exhibit HER2 overexpression, while approximately 50% of HER2-positive breast cancers display positive hormone receptor expression [174,175]. TPBC stands as a distinct subtype characterized by unique tumor cell traits, molecular functions, cellular signaling pathways, therapeutic responsiveness, and biological behavior, diverging from other molecular subtypes [176]. For TPBC patients, both targeted and endocrine therapies are options (Table 1). However, due to the intersection of the HER2 gene and ER-mediated signaling pathways, TPBC features a distinct drug response and resistance mechanism unlike other molecular subtypes [177].

Although the TME significantly influences tumor development and therapeutic responsiveness, cancer cells do not passively respond to TME regulation. Instead, they engage in intricate interactions with the tumor microenvironment, synergistically promoting cancer cell survival, proliferation, and metastasis. Although research on the tumor microenvironment in TNBC has gained momentum, TPBC remains largely underexplored. This review discussed potential TME components relevant to the development of TPBC tumors and resistance to endocrine and trastuzumab therapies (Table 2). Notably, among these components, CAAs and exosomes seem to play pivotal roles in TPBC tumor progression and treatment resistance. However, the intricate nature of TME constituents poses challenges for in vitro studies that strive to replicate the true tumor microenvironment, resulting in limited relevant research. Consequently, future investigations should prioritize the establishment of robust experimental models, utilizing as many TPBC cell lines as possible, to accurately simulate the TPBC tumor microenvironment.

## Figures and Tables

**Table 1 cancers-15-05493-t001:** Clinical trial targeting both HER2 and ER for TPBC.

NCT	Subjects	Treatment	Primary Endpoint
02003209	Terminal TPBC	docetaxel, carboplatin, trastuzumab, and pertuzumab (TCHP) with or without estrogen deprivation (goserelin plus an AI)	N
01817452	Early TPBC	conjugate ado-trastuzumab emtansine (T-DM1) versus trastuzumab with ET	Y
02530424	Early TPBC	CDK4/6 inhibitor palbociclib with trastuzumab, pertuzumab, and fulvestrant	Y
03272477	Early TPBC	ET versus paclitaxel 80 mg/m^2^ weekly plus trastuzumab and pertuzumab	Y
00629278	Early TPBC	AIs vs. tamoxifen or tamoxifen	Y
00878709	Early TPBC	neratinib plus trastuzumab	Y
00022672	Terminal TPBC	anastrozole plus trastuzumab	Y
00073528	Terminal TPBC	letrozole plus lapatinib	Y
01160211	Terminal TPBC	prior trastuzumab and ET	Y

Abbreviations: N, not met; Y, yes.

**Table 2 cancers-15-05493-t002:** Clinical significance and involved mechanisms of tumor stromal cell and non-cellular components in TPBC.

Items	Clinical Significance	Involved Mechanisms	References
CAFs	proliferation and metastasis; tamoxifen resistance; Herceptin resistance	EGFR/ERK/PKC signal; miR-500a-5p/USP28; JAK/STAT3 or PI3K/AKT pathways; GPER/EGFR/ERK signaling and E2 production; PI3K/AKT and MAPK/ERK1/2 pathways IL-6/STAT–3/NF-κB	[39,46,47,48,49,50,51]
TAMs	immune escape and lung metastasis; paclitaxel resistance; tamoxifen resistance	TAM/CXCL1/NF-κB/FOXP3; IL-10/STAT3/bcl-2 signaling pathway; PI3K/AKT/mTOR; NF-κB/STAT3/ERK; EGFR/PI3K/Akt	[67,71,72,73,74]
CAAs	cellular proliferation; promotes EMT; inhibition of adhesion, migration and invasion; endocrine therapy	estrogen receptor (ER) signaling; EGFR/JAK2/HER2; JAK/STAT3 PI3K/AKT; PKM2/PI3K/AKT; LKB1-AMPK-S6K; FGF1/FGFR phosphorylation	[94,95,96,97,98,99,100,109,118]
ASCs	inhibit proliferation; increase breast ball formation, cell fusion, and cell migration	SDF-1α/CXCR4	[127]
MSCs	promote BCSCs self-renewal and cancer cell proliferation; inhibit migration	CCL5/TGF-β/EMT; IL-6 and CXCL7; mTOR/HIF-1α/VEGF	[122,123,125]
TDEs	tamoxifen resistance; trastuzumab resistance	miRNA205-Caspase signaling pathway; lncRNA AGAP2-AS1-ATG10-autophagy reaction; miR-567-autophagy reaction	[141,143,146]

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
