# Peer review of "The Role of the Tumor Microenvironment in Triple-Positive Breast Cancer Progression and Therapeutic Resistance"

_cancers, 2023, doi:10.3390/cancers15225493_

Round 1
Reviewer 1 Report
Comments and Suggestions for Authors
Recommendation: Consider after major revisions
In this review, authors have explored the relationship between critical cellular components and factors in the TPBC microenvironment and the inception, advancement, and therapeutic resistance of breast cancer to provide perspectives on the latest research on TPBC. However, some major issues need to be addressed before this manuscript can be considered for publication.
1. On Page 1 line 29, BRCA should first be introduced before starting off with its abbreviation. Similarly, TNBC should be introduced too.
2. Some schematics in the text should be added for better understanding of the role of TME in TPBC progression; crosstalk between HER2 and ER pathways; etc. Alternatively, some results from reported literature could also be added after obtaining permissions to support the importance of the topic.
3. Page 6 line 302, “Counteracting the effects of leptin through anti-leptin receptor antibody (anti-OBR) and PI3K/AKT signaling inhibitor LY294002 significantly suppresses leptin-induced PKM2 expression, as well as the expression of EMT-related markers.”, authors are suggested to add reference of the study.
4. Page 7 line 348, introduce the abbreviation ‘MSCs’. Please check for abbreviations throughout the manuscript.
5. Page 8 line 374, more clear explanation of the ‘co-culture conditions’ should be presented with reference for better clarity to the readers.
6. Page 10 line 493, tumor-associated adipocytes (CAAs) should be changed to cancer-associated adipocytes (CAAs).
7. As a summary, various components of TME (cellular and non-cellular) along with their roles (cytokines released/ effect seen in literature) could be presented in a tabular form.
8. Discussion should also shed light on lack of 3D culture models in evaluating role of TME in cancer progression under in vitro conditions.
Author Response
Qilu Hospital of Shandong University
26/10/2023
Journal name: Cancers
Manuscript ID: cancers-2641049
Type of manuscript: Review
Title: The role of the tumor microenvironment in triple-positive breast cancer progression and therapeutic resistance
Authors: Qian Pu and Haidong Gao*
Received: 15 Sep 2023
Dear Editor-In-Chief,
I am writing this letter in reference to our submitted article, "Manuscript ID: cancers-2641049 Title: The role of the tumor microenvironment in triple-positive breast cancer progression and therapeutic resistance," to your esteemed journal. Thank you for the opportunity to revise our manuscript. I am submitting the revised manuscript on behalf of all authors for your consideration. We made minor modifications to the original tables; regrettably, we are unable to incorporate additional charts or tables. According to the reviewers’ comments and suggestions, we have tried our best to modify our manuscript to meet the requirements of your journal. All references are relevant to the contents of the manuscript. The red text in this revised version indicates changes to our manuscript within the document. Point-by-point responses is listed below this letter.
COMMENTS AND SUGGESTIONS FOR AUTHORS:
Reviewer #1: Recommendation: Consider after major revisions
In this review, authors have explored the relationship between critical cellular components and factors in the TPBC microenvironment and the inception, advancement, and therapeutic resistance of breast cancer to provide perspectives on the latest research on TPBC. However, some major issues need to be addressed before this manuscript can be considered for publication.
- On Page 1 line 29, BRCA should first be introduced before starting off with its abbreviation. Similarly, TNBC should be introduced too.
Response: The line numbers marked by the reviewer in PDF(cancers-2641049) are not consistent with those in WORD(cancers-2641049), so we have made corresponding modifications in WORD version according to the line numbers marked by the reviewer in PDF(cancers-2641049).
Thank you for your comments. According to the reviewer’s suggestion, we introduced BRCA and TNBC before starting with its abbreviation on Page 1, lines 29 and 42.
- Some schematics in the text should be added for a better understanding of the role of TME in TPBC progression; crosstalk between HER2 and ER pathways; etc. Alternatively, some results from reported literature could also be added after obtaining permissions to support the importance of the topic.
Response: We agree with the reviewer that this is an important question. Recently (2023-05-31) , ,Mark Pegram et al. provided an overview of crosstalk between the ER and HER2 pathways [1].Laboratory research findings have demonstrated that bidirectional crosstalk between the ER and HER2 pathways in breast cancer may lead to the activation activate of the alternative “escape” survival pathways through ER signaling and, eventually, treatment resistance and tumor growth. HER2 overactivation results in the downregulation of ER-regulated transcription and resistance to endocrine therapy. The blockade of HER2 leads to the activation of ER gene transcription via signaling crosstalk pathways as a compensatory mechanism for tumor growth and survival. Inhibition of HER2 and ER pathways is required to achieve effective HER2+/HR+ antitumor activity. In addition, it has been hypothesized that blockade of multiple HER family receptors may be more effective than blockade of HER2 kinase alone, and use of the pan-HER TKI neratinib as part of combination therapy may re-sensitize ER pathways to ET. These results warrant further investigation into the safety and efficacy of dual ER/HER2 blockade in the clinical setting. ; for more details, refer to Page 2, lines 63-69.When one pathway is blocked by chemotherapy or hormone therapy, the other is subsequently upregulated . Mark Pegram et al. pointed out that blocking multiple HER family receptors may be more effective than blocking the HER 2 kinase alone, and using pan-HER TKI neritinib as part of the combination therapy may re-sensitize the ER pathway to endocrine therapy. The schematic drawings in the reference are as follows:
Fig. estrogen receptor (ER) and human epidermal growth factor receptor 2 (HER2) signaling crosstalk[1]
Gray shading of the PI3K/AKT/mTOR and RAS/RAF/MEK/ERK signaling pathways in panels a and b indicate downregulation of these pathways. AI aromatase inhibitor, ER estrogen receptor,ERK extracellular signal-regulated kinase, HER2 human epidermal growth factor receptor 2, HR hormone receptor, MEK mitrogen-activatedprotein kinase kinase, mTOR mammalian target of rapamycin, P13K phosphatidylinositol 3-kinase, RAF rapidly accelerated fibrosarcoma, SERDselective estrogen receptor degrader, SERM selective estrogen receptor modulator, TKI tyrosine kinase inhibitor.
- Page 6 line 302, “Counteracting the effects of leptin through anti-leptin receptor antibody (anti-OBR) and PI3K/AKT signaling inhibitor LY294002 significantly suppresses leptin-induced PKM2 expression, as well as the expression of EMT-related markers.”,authors are suggested to add reference of the study.
Response: Thank you for your comments. The reference of the study is Wei, L. et al., Leptin promotes epithelial-mesenchymal transition of breast cancer via the upregulation of pyruvate kinase M2. J Exp Clin Cancer Res, 2016. 35(1): p. 166. PMID: 27769315.Page 7, line 320 and Page 17, line 807.
- Page 7 line 348, introduce the abbreviation ‘MSCs’. Please check for abbreviations throughout the manuscript.
Response: Thank you for your comments. I apologize for our mistakes. The mistake in PDF (cancers-2641049) Page 7 line 348 has been corrected in WORD (cancers-2641049), Page 8, line 365.
- Page 8 line 374, more clearexplanation of the ‘co-culture conditions’ should be presented with reference for better clarity to the readers.
Response: We agree with the reviewer that this is an important question. The reference is No. 126 Page 8 line 398( Page 18 line 861). The tumor cells were either directly cocultured with AT-MSCs or exposed to MSCs-conditioned medium (MSC-CM). AT-MSCs were isolated and characterized by immunophenotype and differentiation potential, as previously described. The AT-MSCs were expanded in low glucose (1.0 g/L) DMEM supplemented with 10% HyClone® AdvanceSTEM™ (Thermo Scientific) and antibiotic/antimycotic mix (10.000 IU/mL penicillin, 5 μg/mL streptomycin, 2 mM glutamine, and 2.5 μg/mL amphotericin).
- Page 10 line 493, tumor-associated adipocytes (CAAs) should be changed to cancer-associated adipocytes (CAAs).
Response: Thank you for your comments. The mistake in PDF (cancers-2641049) Page 10 line 493 has been corrected in WORD (cancers-2641049) Page 10, line 513.
7.As a summary, various components of TME (cellular and non-cellular) along with their roles (cytokines released/effect seen in literature) could be presented in a tabular form.
Response: Thanks for your suggestion. We have supplemented the original form content.
- Discussion should also shed light on the lack of 3D culture models in evaluating the role of TME in cancer progression under in vitro conditions.
Response: Thank you for your comments. We agree with the reviewer that this is an important question. Establishing a three-dimensional cell culture system in vitro can more closely mimic the real in vivo environment. For instance, sandwich culture and organoid culture systems have overcome many limitations of 2D plane culture. However, they still have inevitable drawbacks: the cells in the sandwich culture system still grow in the plane, and the two-dimensional network structure collapses over time, limiting its suitability for long-term pharmacological studies [2]. The establishment process of organoid culture systems is complex, and the culture system requires expensive growth factors and small-molecule compounds. Consequently, the construction and mating processes are expensive, and the construction of an organoid culture system is highly dependent on parental tumor proliferation and has a low construction success rate of only 26% [3].The patient-derived tumor Xenograft (PDX) model established an in vitro tumor research model from another perspective, in which transplantation of tumor tissue into immunodeficient animals can greatly retain the characteristics of parental tumor tissue both at the gene level and histopathology [4, 5]. However, there is a high dependence on the proliferation ability of parental tumor tissue, and the success rate of in vitro modeling of breast cancer does not exceed 50% [6, 7]. Our most recent study is an experiment using the PDX model [8]. In summary, although the corresponding tumor models can be established in vitro to some extent, problems such as high economic cost, low construction success rate, and short cell survival time weaken the intention of the above model for individualized drug screening and do not have clinical universality [9, 10].
Currently, there are few studies that utilize the 3D culture model to assess the role of TME in the in vitro conditions in TPBC progression. Recently, A team led by Professor Mengsu Yang from the City University of Hong Kong reviewed 3D bionic models for in vitro reconstruction of tumor microenvironments: spheres, organoids, and tumor chips [11]. This article reviews recent advances in in vitro platforms for cancer modeling and preclinical drug screening, provides a brief overview of the tumor microenvironment (TME) components and its role in tumor progression, and highlights the potential for microspheres/organoids combined with microfluidic techniques to simulate tumors in vivo better. We believe that this may provide new ideas for future research. In our manuscript, we have collated and analyzed the roles of cellular and noncellular components in TPBC tumor microenvironment. Additionally, as warranted, we will delve deeper into the 3D culture model to evaluate the role of TME in TPBC progression under in vitro conditions.
Thank you again for your positive comments and valuable suggestions to improve our manuscript quality. If there are any other modifications we could make, we are receptive to your feedback and appreciate your help. We hope that you find our responses and modifications satisfactory and that the manuscript is now acceptable for publication. Thank you for your help.
Yours Sincerely,
Haidong Gao
Department of General Surgery
Qilu Hospital of Shandong University
F12 Breast Surgery Ward, Qilu Hospital of Shandong University (Qingdao), No. 758, Hefei Road, Shibei District, Qingdao, Shandong, 266000, PR China.
E-mail address: haidonggao@sdu.edu.cn
Phone: 18561811668
References
- Pegram, M., C. Jackisch, and S.R.D. Johnston, Estrogen/HER2 receptor crosstalk in breast cancer: combination therapies to improve outcomes for patients with hormone receptor-positive/HER2-positive breast cancer.NPJ Breast Cancer, 2023. 9(1): p. 45.
- Ravi, M., et al., 3D cell culture systems: advantages and applications.J Cell Physiol, 2015. 230(1): p. 16-26.
- Nuciforo, S., et al., Organoid Models of Human Liver Cancers Derived from Tumor Needle Biopsies.Cell Rep, 2018. 24(5): p. 1363-1376.
- Fong, E.L.S., et al., Generation of matched patient-derived xenograft in vitro-in vivo models using 3D macroporous hydrogels for the study of liver cancer.Biomaterials, 2018. 159: p. 229-240.
- Sun, L., et al., Modelling liver cancer initiation with organoids derived from directly reprogrammed human hepatocytes.Nat Cell Biol, 2019. 21(8): p. 1015-1026.
- Murayama, T. and N. Gotoh, Patient-Derived Xenograft Models of Breast Cancer and Their Application.Cells, 2019. 8(6).
- Aparicio, S., M. Hidalgo, and A.L. Kung, Examining the utility of patient-derived xenograft mouse models.Nat Rev Cancer, 2015. 15(5): p. 311-6.
- Qiao, L., et al., Self-destructive Copper Carriers Induce Pyroptosis And Cuproptosis for Efficient Tumor Immunotherapy Against Dormant And Recurrent Tumors.Adv Mater, 2023: p. e2308241.
- Cheung, P.F., et al., Comprehensive characterization of the patient-derived xenograft and the paralleled primary hepatocellular carcinoma cell line.Cancer Cell Int, 2016. 16: p. 41.
- Drost, J. and H. Clevers, Organoids in cancer research.Nat Rev Cancer, 2018. 18(7): p. 407-418.
- Li, W., et al., 3D Biomimetic Models to Reconstitute Tumor Microenvironment In Vitro: Spheroids, Organoids, and Tumor-on-a-Chip.Adv Healthc Mater, 2023. 12(18): p. e2202609.

Reviewer 2 Report
Comments and Suggestions for Authors
In this report, Pu & Gao attempt to review the complex role of tumor microenvironment in TPBC, discussing the latest research in this context. This is an important point of view, since it’s now clear that breast carcinogenesis is related not only to the intrinsic features of cancer cells, but it is also strongly dependent on surrounding microenviromental factors. Moreover, TPBC accounts for approximately 10% to 15% of all breast cancers, and studies focused on this subtype are still limited. The authors sufficiently described the state of the art and reported the fundamental concepts. Overall, the review paper is well written, with a coherent flow of information, allowing an easy comprehension of the intricate dynamics within TME. However, the review would benefit from some implementation:
- - The literature searching strategy was not reported;
- - Few schematic figures are mandatory to summarize the conclusions;
- - It is important to associate sections 1 and 2 with specific tables reporting the main findings of studies on TME and TPBC, subdivided for ‘in vitro’ and ‘in vivo’ experimental evidences and epidemiological and ‘ex vivo’ clinical evidences;
- - It is limitative to talk only of TDEs. The readers should have a more comprehensive overview of the involvement of extracellular vesicles derived from tumor epithelial cells and other cells within TME;
- - It lacks a section focused on present and future pharmaceutical strategies that could interfere with TME action and thereby prevent/delay TPBC initiation and progression. Again, a table should be associated, at least for the ongoing clinical trials;
- -- The current research gaps and the challenges in the field are not discussed, and surely, they derseve space in the conclusion section as an additional paragraph;
- - Few typos need to be edited (i.e. 1.4 Mesenchymal Stem Cells, 2. Non-cellular components in the TME of TPBC in bold, etc).
Comments on the Quality of English Language
Minor editing of English language required.
Author Response
Qilu Hospital of Shandong University
26/10/2023
Journal name: Cancers
Manuscript ID: cancers-2641049
Type of manuscript: Review
Title: The role of the tumor microenvironment in triple-positive breast cancer progression and therapeutic resistance
Authors: Qian Pu and Haidong Gao*
Received: 15 Sep 2023
Dear Editor-In-Chief,
I am writing this letter in reference to our submitted article, "Manuscript ID: cancers-2641049 Title: The role of the tumor microenvironment in triple-positive breast cancer progression and therapeutic resistance," to your esteemed journal. Thank you for the opportunity to revise our manuscript. I am submitting the revised manuscript on behalf of all authors for your consideration. We made minor modifications to the original tables; regrettably, we are unable to incorporate additional charts or tables. According to the reviewers’ comments and suggestions, we have tried our best to modify our manuscript to meet the requirements of your journal. All references are relevant to the contents of the manuscript. The red text in this revised version indicates changes to our manuscript within the document. Point-by-point responses is listed below this letter.
COMMENTS AND SUGGESTIONS FOR AUTHORS:
Reviewer #2:Comments and Suggestions for Authors
In this report, Pu & Gao attempt to review the complex role of tumor microenvironment in TPBC, discussing the latest research in this context. This is an important point of view, since it’s now clear that breast carcinogenesis is related not only to the intrinsic features of cancer cells, but it is also strongly dependent on surrounding microenviromental factors. Moreover, TPBC accounts for approximately 10% to 15% of all breast cancers, and studies focused on this subtype are still limited. The authors sufficiently described the state of the art and reported the fundamental concepts. Overall, the review paper is well written, with a coherent flow of information, allowing an easy comprehension of the intricate dynamics within TME. However, the review would benefit from some implementation:
- - The literature searching strategy was not reported;
Response: Thank you for your comments. The keywords "tumor microenvironment" and "triple-positive breast cancer" were selected. Articles including triple-positive breast cancer progression and therapeutic resistance were collected from PubMed, Springer, and OVID databases, and the relevant data acquired were analyzed. We did not restrict our search to the study design or publication date status.
- - Few schematic figures are mandatory to summarize the conclusions;
Response: Thank you for your comments. While our aim remains to publish this manuscript in your esteemed journal, we have found very few meaningful things. We postulate that fat cells or related cellular components warrant further investigation in the context of breast cancer. The intricacies of triple-positive breast cancer remain elusive, and we regret that our perspective remains incomplete. We are grateful for your feedback and intend to continue our research endeavors. We aspire to consolidate our findings in the future to produce representations that meet both our standards and expectations.
- - It is important to associate sections 1 and 2 with specific tables reporting the main findings of studies on TME and TPBC, subdivided for ‘in vitro’ and ‘in vivo’ experimental evidences and epidemiological and ‘ex vivo’ clinical evidences;
Response: Thank you for your comments. The references involve in vitro experiments using experimental studies of cell lines.
- - It is limitative to talk only of TDEs. The readers should have a more comprehensive overview of the involvement of extracellular vesicles derived from tumor epithelial cells and other cells within TME;
Response: Thank you for your comments. We agree with the reviewer that this is an important question.
Extracellular Vesicles (EV)
Microvesicles and exosome systems are called extracellular vesicles, according to the Intracellular Journal of Extracellular Vesicles criteria [1, 2].A recent study elucidated the heterogeneity of extracellular nanoparticles and defined three distinct subpopulations, namely small exosomes (Exo-S), large exosomes (Exo-L), and exoparticles (Exomeres), collectively known as extracellular vesicles and granules (EVP) [3].Many studies have revealed that breast cancer-derived EVs are enriched with various cancer-related tumor markers, such as HER [4, 5]and EGFR[6].
Exosomes range from 30 to 150 nm and originate from the endocytic pathway; tumor vesicle diameters of 1~10 µm are produced by tumor cells; and exoparticle diameters of 100~1 000 nm are directly released from the plasma membrane. Based on this, this review used the term "extracellular vesicles" to initially search the extracellular vesicle database Vesiclepedia (htp://microvesicles.org/index.html) for all vesicles derived from 30 nm to 10 µm tumors. We felt high exosome and triple-positive breast cancer progression and treatment resistance; thus, our discussion was solely centered on exosomes. Thank you very much for your question. Our future work will also focus on the role of EV in triple-positive breast cancer, and we look forward to improving our findings.
- - It lacks a section focused on present and future pharmaceutical strategies that could interfere with TME action and thereby prevent/delay TPBC initiation and progression. Again, a table should be associated, at least for the ongoing clinical trials
Response: Thank you for your comments. We believe that this matter includes two small issues: 1) strategies that discuss current and future drugs that may interfere with the effects of TME and thus prevent/delay TPBC initiation and progression; 2) requiring a table to present the currently ongoing clinical trials.
To date, combined blockade of ER/HER 2 signaling using drugs directly targeting ER (e.g., selective estrogen receptor degraders [SERDs], selective estrogen receptor modulators [SERMs], AIs) and HER 2 (e.g., monoclonal antibodies or small molecule TKIs) has been revealed to be highly effective in preclinical models [7].Recent studies have also evaluated neratinib in combination with other targeted agents to overcome treatment resistance in preclinical models of breast cancer [8-10]. In a human-mouse model of ER +/HER 2 + breast cancer, adjuvant paclitaxel plus trastuzumab±pertuzumab for four weeks, followed by fulvestrant plus "extended adjuvant therapy" nelatinib was associated with maintaining complete remission (CR), while treated subjects were rapidly relapsed with prolonged fulvestrant alone [8]. Indeed, more research is required to identify subsets of patients that will receive the best clinical benefit from currently available treatments, as well as novel combination treatment strategies and novel agents with downstream effects that may improve clinical outcomes.
Recently, Mark Pegram et al. summarized data from recently published and ongoing clinical trials and discussed the clinical implications of targeted treatment of HR+/HER2+ breast cancer[11]This review has been a very comprehensive summary of currently ongoing clinical trials, which we simply added. The ongoing DETECT V/CHEVENDO study comparing the safety and efficacy of ribociclib combined with trastuzumab+pertuzumab vs. endocrine therapy or chemotherapy is also a pending publication (S Krause et al. Abstract OT2-07-01: DETECT V/CHEVENDO–Comparison of dual HER2-targeted therapy with trastuzumab plus pertuzumab combined with chemo-or endocrine therapy and CDK4/6 inhibition in patients with HER2-positive and hormone-receptor-positive metastatic breast cancer. Cancer Res. 2019. 79 (4_Supplement): OT2-07-01). Please allow us not to repeat the tables of the clinical trials.
- -- The current research gaps and the challenges in the field are not discussed, and surely, they derseve space in the conclusion section as an additional paragraph
Response: Thank you for your comments. We agree with the reviewer that this is an important question.
We believe that the search for more effective predictive biomarkers of triple-positive breast cancer may be a hotspot for future research. Serial clinical studies have started to attempt the preoperative use of molecular profiles to guide decision-making for neoadjuvant therapy in breast cancer. PerELISA [12]and PAMELA [13]et al. found that using PAM50 analysis of three-positive early breast cancer, the pCR rate enriched by HER 2 was higher than that of other endogenous subtypes (41%~54% vs. 13.8%~28%). Besides, the I-SPY study used comprehensive data sets (transcriptome, proteome, and clinical response data) with the therapeutic efficacy of 10 different therapeutic drugs/strategies. For reclassification of breast cancer with the best efficacy, the response predictive subtype was constructed (response predictive subtypes, RPS-5) pattern; the results indicated that the pCR rate could increase from 51% to 67%. This suggests that RPS-5 is important for predicting the efficacy of therapeutic agents and has application prospects[14]. Additionally, some studies have analyzed the correlation between pCR and multigene testing to determine tumor biological characteristics, pCR rate, and survival prognosis. Among them, the most studied is HER2DX, which refers to the comprehensive evaluation score of 27 genes combined with clinicopathological characteristics (tumor size, lymph node status, etc.) and holds prognostic value as well as the predictive potential for pCR[15]. The results also suggest a strong correlation between immune characteristics and pCR in HER 2-positive breast cancer, underscoring a potential avenue for future research [16].. Here, our review delves into cellular and non-cellular components in the TME, and if necessary, we have added this biological prediction and subtyping to the Discussion section.
- - Few typos need to be edited (i.e. 1.4 Mesenchymal Stem Cells, 2. Non-cellular components in the TME of TPBC in bold, etc).
Response: Thank you for your comments. The mistake has been corrected in WORD (cancers-2641049) Page8, line 365 and Page 8 line 414.
Thank you again for your positive comments and valuable suggestions to improve our manuscript quality. If there are any other modifications we could make, we are receptive to your feedback and appreciate your help. We hope that you find our responses and modifications satisfactory and that the manuscript is now acceptable for publication. Thank you for your help.
Yours Sincerely,
Haidong Gao
Department of General Surgery
Qilu Hospital of Shandong University
F12 Breast Surgery Ward, Qilu Hospital of Shandong University (Qingdao), No. 758, Hefei Road, Shibei District, Qingdao, Shandong, 266000, PR China.
E-mail address: haidonggao@sdu.edu.cn
Phone: 18561811668
References
- Thery, C., et al., Minimal information for studies of extracellular vesicles 2018 (MISEV2018): a position statement of the International Society for Extracellular Vesicles and update of the MISEV2014 guidelines.J Extracell Vesicles, 2018. 7(1): p. 1535750.
- Shao, H., et al., New Technologies for Analysis of Extracellular Vesicles.Chem Rev, 2018. 118(4): p. 1917-1950.
- Hoshino, A., et al., Extracellular Vesicle and Particle Biomarkers Define Multiple Human Cancers.Cell, 2020. 182(4): p. 1044-1061 e18.
- Fang, S., et al., Clinical application of a microfluidic chip for immunocapture and quantification of circulating exosomes to assist breast cancer diagnosis and molecular classification.PLoS One, 2017. 12(4): p. e0175050.
- Amorim, M., et al., The overexpression of a single oncogene (ERBB2/HER2) alters the proteomic landscape of extracellular vesicles.Proteomics, 2014. 14(12): p. 1472-9.
- Ortega, F.G., et al., EGFR detection in extracellular vesicles of breast cancer patients through immunosensor based on silica-chitosan nanoplatform.Talanta, 2019. 194: p. 243-252.
- Croessmann, S., et al., Combined Blockade of Activating ERBB2 Mutations and ER Results in Synthetic Lethality of ER+/HER2 Mutant Breast Cancer.Clin Cancer Res, 2019. 25(1): p. 277-289.
- Sudhan, D.R., et al., Extended Adjuvant Therapy with Neratinib Plus Fulvestrant Blocks ER/HER2 Crosstalk and Maintains Complete Responses of ER(+)/HER2(+) Breast Cancers: Implications to the ExteNET Trial.Clin Cancer Res, 2019. 25(2): p. 771-783.
- Ribas, R., et al., Targeting tumour re-wiring by triple blockade of mTORC1, epidermal growth factor, and oestrogen receptor signalling pathways in endocrine-resistant breast cancer.Breast Cancer Res, 2018. 20(1): p. 44.
- Nayar, U., et al., Acquired HER2 mutations in ER(+) metastatic breast cancer confer resistance to estrogen receptor-directed therapies.Nat Genet, 2019. 51(2): p. 207-216.
- Pegram, M., C. Jackisch, and S.R.D. Johnston, Estrogen/HER2 receptor crosstalk in breast cancer: combination therapies to improve outcomes for patients with hormone receptor-positive/HER2-positive breast cancer.NPJ Breast Cancer, 2023. 9(1): p. 45.
- Guarneri, V., et al., De-escalated therapy for HR+/HER2+ breast cancer patients with Ki67 response after 2-week letrozole: results of the PerELISA neoadjuvant study.Ann Oncol, 2019. 30(6): p. 921-926.
- Llombart-Cussac, A., et al., HER2-enriched subtype as a predictor of pathological complete response following trastuzumab and lapatinib without chemotherapy in early-stage HER2-positive breast cancer (PAMELA): an open-label, single-group, multicentre, phase 2 trial.Lancet Oncol, 2017. 18(4): p. 545-554.
- Wolf, D.M., et al., Redefining breast cancer subtypes to guide treatment prioritization and maximize response: Predictive biomarkers across 10 cancer therapies.Cancer Cell, 2022. 40(6): p. 609-623 e6.
- Prat, A., et al., Development and validation of the new HER2DX assay for predicting pathological response and survival outcome in early-stage HER2-positive breast cancer.EBioMedicine, 2022. 75: p. 103801.
- Fernandez-Martinez, A., et al., Prognostic and Predictive Value of Immune-Related Gene Expression Signatures vs Tumor-Infiltrating Lymphocytes in Early-Stage ERBB2/HER2-Positive Breast Cancer: A Correlative Analysis of the CALGB 40601 and PAMELA Trials.JAMA Oncol, 2023. 9(4): p. 490-499.
Reviewer 3 Report
Comments and Suggestions for Authors
Thank you for submitting this interesting review article. However the manuscript was too long and need to be more focus and reorganised.
I would suggest to include the following sections:
1- HER2/ER signalling cross talk and thier interaction with micro environment. More focus on its clinical implication, therapy resistance and opportunities for finding novel therapeutic target is needed.
2- Endothelial cell and angiogenesis with more focus on its therapeutic implication regarding current therapeutic resistance and new therapeutic strategies.
3- Novel methods and technologies that could effectively facilitate our understanding of tumour/ microenvironment cross talk.
Comments on the Quality of English Language
The English language was fine.
Author Response
Qilu Hospital of Shandong University
26/10/2023
Journal name: Cancers
Manuscript ID: cancers-2641049
Type of manuscript: Review
Title: The role of the tumor microenvironment in triple-positive breast cancer progression and therapeutic resistance
Authors: Qian Pu and Haidong Gao*
Received: 15 Sep 2023
Dear Editor-In-Chief,
I am writing this letter in reference to our submitted article, "Manuscript ID: cancers-2641049 Title: The role of the tumor microenvironment in triple-positive breast cancer progression and therapeutic resistance," to your esteemed journal. Thank you for the opportunity to revise our manuscript. I am submitting the revised manuscript on behalf of all authors for your consideration. We made minor modifications to the original tables; regrettably, we are unable to incorporate additional charts or tables. According to the reviewers’ comments and suggestions, we have tried our best to modify our manuscript to meet the requirements of your journal. All references are relevant to the contents of the manuscript. Point-by-point responses are listed below this letter.
COMMENTS AND SUGGESTIONS FOR AUTHORS:
Reviewer #3:Comments and Suggestions for Authors
Thank you for submitting this interesting review article.However the manuscript was too long and need to be more focus and reorganised.
I would suggest to include the following sections:
- HER2/ER signalling crosstalk and thier interaction with micro More focus on its clinical implication, therapy resistance and opportunities for finding novel therapeutic target is needed.
Response We agree with the reviewer that this is an important question. Progress is relatively slow in patients with triple-positive breast cancer; endocrine drugs combined with double targets are preferred, and ADC drugs can be considered after endocrine therapy resistance. The combination of CDK4/6 inhibitors and HER2-targeting drugs has recently been continuously explored. The PATRICIA study explored the efficacy of trastuzumab + piperocillin + letrozole in posterior line therapy (2–4 lines) for HER2-positive advanced breast cancer, showing that the median PFS could reach 12.4 months [1](Ciruelos E et al. Abstract P5-20-19: PAM50 intrinsic subtype predicts survival outcome in HER2-positive/hormone receptor-positive metastatic breast cancer treated with palbociclib and trastuzumab: a correlative analysis of the PATRICIA (SOLTI 13-03) trial: AACR; 2018). Another Phase III PATINA trial involving piperoxiril is investigating the efficacy of Piperoxiril or placebo besides trastuzumab, pertuzumab, and endocrine therapy maintenance therapy after four to eight cycles of induction chemotherapy and HER2-targeted therapy in metastatic breast cancer. The results are pending (O Metzger et al. Abstract OT3-02-07: PATINA: A randomized, open-label, phase III trial to evaluate the efficacy and safety of palbociclib + anti-HER2 therapy + endocrine therapy (ET) vs. anti-HER2 therapy + ET after induction treatment for hormone receptor-positive (HR+)/HER2-positive metastatic breast cancer (MBC) Cancer Res (2019) 79 (4_Supplement): OT3-02-07). The ongoing DETECT V/CHEVENDO study comparing the safety and efficacy of ribociclib combined with trastuzumab + pertuzumab vs. endocrine therapy or chemotherapy is also pending publication (S Krause et al. Abstract OT2-07-01: DETECT V/CHEVENDO–Comparison of dual HER2-targeted therapy with trastuzumab plus pertuzumab combined with chemo-or endocrine therapy and CDK4/6 inhibition in patients with HER2-positive and hormone-receptor-positive metastatic breast cancer. Cancer Res. 2019. 79 (4_Supplement): OT2-07-01).
Moreover, HR+/HER2+ breast cancer is less sensitive to endocrine therapy than HR+/HER2- breast cancer at the mechanistic level because HER2 signaling activates CDK4/6 by increasing CCND1 expression or cyclin D1 stability[2]. When HER2 pathway is blocked, ER pathway can be bypass-activated to stimulate tumor proliferation further and produce drug resistance. Therefore, inhibition of ER and HER2 pathways in HR+/HER2+ breast cancer can improve clinical benefit rates[3, 4].2- Endothelial cell and angiogenesis with more focus on its therapeutic implication regarding current therapeutic resistance and new therapeutic strategies.
Response:Thank you for your comments. We agree with the reviewer that this is an important question.
Tumor endothelial cells (TECs)
At present, there is little research on endothelial cells in the microenvironment of triple-positive breast cancer tumors; therefore, we do not present relevant content in this paper. Long-term TME stimulation causes abnormalities in the morphology, phenotype, function, and gene expression of normal vascular endothelial cells, eventually transforming into tumor endothelial cells (TECs). Unlike normal endothelial cells, TECs have many channels and holes in the tube wall, resulting in high permeability and an incomplete or missing basement membrane between cells. These characteristics are conducive to tumor cells to penetrate the blood vessel wall and complete distant metastasis. Additionally, the vascular endothelial growth factor (VEGF) receptor-1 and VEGF receptor-2 expression levels were up-regulated compared to normal endothelial cells, resulting in increased VEGF sensitivity. It is in an active state of high proliferation and high metastasis. Therefore, TECs can promote tumor neovascularization formation and affect cancer cell growth and metastasis. Recent studies have discovered that tumor endothelial cells can also release vascular endothelial-derived factors (collectively known as angiocrine), regulating angiogenesis and tumor growth. Ghiabi et al. reported that Jag1 ligand overexpression of TECs after contact with breast cancer cells activates the notch signaling pathway in TNBC cell lines MDA-MB231 and Luminal cell lines MCF7, forming a niche that promotes cancer cell growth and metastasis[5]. Lee et al. proved that the interaction between vascular endothelial cells and tumor cells can cause the cancer cells to acquire stem-like properties and promote epithelial-mesenchymal transformation [6]. The current research on endothelial cells in triple-positive breast cancer is notably limited, and we will continue to pay further attention.
3- Novel methods and technologies that could effectively facilitate our understanding of tumour/ microenvironment cross talk.
Response: T-DXd, a novel ADC drug, is not completely dependent on the blocking effect of the HER 2 signaling pathway. Its anti-tumor killing has a high DAR and bystander effect, and its efficacy is unlimited by ER/PR status. Therefore, T-DXd treatment in three-positive breast cancers has some theoretical support. The DB03 study’s subgroup analysis results revealed that the median PFS in HR-positive patients was 22.4 months, representing a 62% lower risk of progression than T-DM1 [7].The results partly validate the efficacy of T-DXd for triple-positive breast cancer. Additionally, a combined approach targeting other pathways is being explored, such as a clinical phase III study of PI3K inhibitor Alpelisib combined with trastuzumab and peruzumab for triple-positive breast cancer [8]. The publication of the research results may provide a specific evidence-based basis for applying PI3K inhibitors in triple-positive breast cancer and broaden the thinking for the clinical exploration of new pan-PI3K/PIK3CA inhibitors in development.
Comments on the Quality of English Language
The English language was fine.
Thank you for your comments; your comments are a great encouragement to my article.
Thank you again for your positive comments and valuable suggestions to improve our manuscript quality. If there are any other modifications we could make, we are receptive to your feedback and appreciate your help. We hope that you find our responses and modifications satisfactory and that the manuscript is now acceptable for publication. Thank you for your help.
Yours Sincerely,
Haidong Gao
Department of General Surgery
Qilu Hospital of Shandong University
F12 Breast Surgery Ward, Qilu Hospital of Shandong University (Qingdao), No. 758, Hefei Road, Shibei District, Qingdao, Shandong, 266000, PR China.
E-mail address: haidonggao@sdu.edu.cn
Phone: 18561811668
References
- Tolaney, S.M., et al., Abemaciclib plus trastuzumab with or without fulvestrant versus trastuzumab plus standard-of-care chemotherapy in women with hormone receptor-positive, HER2-positive advanced breast cancer (monarcHER): a randomised, open-label, phase 2 trial.Lancet Oncol, 2020. 21(6): p. 763-775.
- Choi, Y.J., et al., The requirement for cyclin D function in tumor maintenance.Cancer Cell, 2012. 22(4): p. 438-51.
- Llombart-Cussac, A., et al., HER2-enriched subtype as a predictor of pathological complete response following trastuzumab and lapatinib without chemotherapy in early-stage HER2-positive breast cancer (PAMELA): an open-label, single-group, multicentre, phase 2 trial.Lancet Oncol, 2017. 18(4): p. 545-554.
- Rimawi, M., et al., First-Line Trastuzumab Plus an Aromatase Inhibitor, With or Without Pertuzumab, in Human Epidermal Growth Factor Receptor 2-Positive and Hormone Receptor-Positive Metastatic or Locally Advanced Breast Cancer (PERTAIN): A Randomized, Open-Label Phase II Trial.J Clin Oncol, 2018. 36(28): p. 2826-2835.
- Ghiabi, P., et al., Endothelial cells provide a notch-dependent pro-tumoral niche for enhancing breast cancer survival, stemness and pro-metastatic properties.PLoS One, 2014. 9(11): p. e112424.
- Lee, E., N.B. Pandey, and A.S. Popel, Crosstalk between cancer cells and blood endothelial and lymphatic endothelial cells in tumour and organ microenvironment.Expert Rev Mol Med, 2015. 17: p. e3.
- Cortes, J., et al., Trastuzumab Deruxtecan versus Trastuzumab Emtansine for Breast Cancer.N Engl J Med, 2022. 386(12): p. 1143-1154.
- Andre, F., et al., Alpelisib plus fulvestrant for PIK3CA-mutated, hormone receptor-positive, human epidermal growth factor receptor-2-negative advanced breast cancer: final overall survival results from SOLAR-1.Ann Oncol, 2021. 32(2): p. 208-217.
Round 2
Reviewer 1 Report
Comments and Suggestions for Authors
The authors have answered the concerns satisfactorily. Thus, the manuscript can be accepted for publication in its present form.
Reviewer 3 Report
Comments and Suggestions for Authors
Thank you for submitting this outstanding revised manuscript which comprehensively addressed all my previous comments. I would like to congratulate the author and coauthors for this manuscript that I believe it would significantly add to the literature and open new avenues for understanding triple positive breast cancer.